# The Association between the L3 Skeletal Muscle Index Derived from Computed Tomography and Clinical Outcomes in Patients with Urinary Tract Infection in the Emergency Department

**DOI:** 10.3390/jcm12155024

**Published:** 2023-07-31

**Authors:** Jinjoo An, Seung Pill Choi, Jae Hun Oh, Jong Ho Zhu, Sung Wook Kim, Soo Hyun Kim

**Affiliations:** 1Department of Emergency Medicine, College of Medicine, The Catholic University of Korea, Seoul 06591, Republic of Korea; 21900454a@gmail.com; 2Department of Emergency, Eunpyeong St. Mary’s Hospital, College of Medicine, The Catholic University of Korea, Seoul 03312, Republic of Korea; emvic98@catholic.ac.kr (S.P.C.); emojh@catholic.ac.kr (J.H.O.); kingmonst2r@gmail.com (J.H.Z.); mdkaptain@naver.com (S.W.K.)

**Keywords:** urinary tract infections, critical care outcomes, skeletal muscle index, computed tomography

## Abstract

The occurrence of a critical event during a urinary tract infection (UTI) can have a significant impact on mortality. This study aimed to investigate the association between the skeletal muscle index (SMI) and critical events in patients with a UTI. From April 2019 to March 2022, a total of 478 patients who met the diagnostic criteria of a UTI and underwent an abdominal CT were included in this study. Multivariate binary logistic regression analysis was used to assess independent predictors of critical events. The primary outcome was any critical event, defined as the initiation of dialysis, invasive ventilation, initiation of vasoactive medications, cardiac arrest, or death. The UTI patients were divided into two groups: those with a low SMI (n = 93) and those with a high SMI (n = 385). In multivariate analysis, a low SMI, diabetes mellitus, altered mentality, lactate levels, and creatinine levels were identified as significant predictors of critical events. A low SMI is an independent factor associated with the occurrence of critical events in UTI patients during hospitalization. Patients with a low SMI, indicating muscle wasting, may have less resilience to infections and a higher risk of experiencing severe complications. Considering the SMI along with other clinical factors can help health care providers assess and manage UTI patients.

## 1. Introduction

The occurrence of a critical event during a urinary tract infection (UTI) can have a significant impact on mortality. UTIs are a common cause of sepsis in hospitals. There are over one million emergency department visits and 100,000 admissions for UTIs annually in the United States. Complicated UTIs (cUTIs), although a fraction of the total UTI volume, costs the health care system over USD 3.5 billion per year [1]. A critical event refers to a severe complication that can arise from a UTI, such as sepsis or kidney failure. These events can increase the risk of mortality in patients with UTIs, especially if they are not promptly recognized and treated. Sepsis, in particular, is a life-threatening condition that can occur when a UTI progresses and spreads to the bloodstream, causing a systemic infection. Sepsis can cause organ failure and can be fatal if not promptly treated. Besides sepsis, other critical events that can arise from a UTI include kidney failure, urinary obstruction, and abscess formation. These events can further increase the risk of mortality, especially in patients with pre-existing medical conditions or compromised immune systems. The prompt recognition and treatment of UTIs and any associated critical events are essential for reducing the risk of mortality. Patients with UTIs should promptly seek medical attention, especially if they develop symptoms such as fever, chills, or severe pain. Timely treatment with appropriate antibiotics and supportive care can help prevent complications and improve outcomes.

The skeletal muscle index (SMI) is a measure of the amount of muscle mass in the body relative to height and weight. Skeletal muscle assessments at the L3 vertebra level have a significant correlation with total body muscle measurements. Low-for-age skeletal muscle mass and function, sarcopenia, is associated with physical frailty and an increased risk of morbidity due to medical comorbidities. Several papers suggested a correlation between skeletal muscle mass and such conditions as in-hospital mortality in community-acquired pneumonia patients, poor functional outcome in patients with acute ischemic stroke, major adverse cardiovascular events in patients with coronary artery disease, and outcomes for trauma patients [2,3,4].

While there is no direct link between the SMI and the risk of critical events in patients with UTIs, the SMI can indirectly impact a patient’s overall health and resilience to infections. A low SMI, or muscle wasting, is a common condition in elderly and chronically ill patients, and it is associated with an increased risk of infection and poor outcomes. Muscle wasting can lead to reduced mobility and strength, which can increase the risk of falls, pressure sores, and other complications that can worsen the outcomes of a UTI. In addition, muscle wasting can be a marker of underlying health conditions that can increase the risk of critical events in patients with UTIs, such as malnutrition, chronic kidney disease, and diabetes. These conditions can weaken the immune system, impair the body’s ability to fight infections, and increase the risk of sepsis and other severe complications. Therefore, while the SMI may not be directly linked to critical events in a UTI, it can be an important indicator of a patient’s overall health status and risk of complications. Health care providers should consider the SMI and other factors when assessing and managing UTIs in patients, especially those with underlying health conditions.

## 2. Materials and Methods

### 2.1. Study Design

This retrospective and observational study included a consecutive cohort of patients admitted to the ED of a hospital located in an urban area in Seoul, South Korea. The institutional review board approved this study (PC22RISI0266). Given its retrospective nature, the requirement for informed consent was waived.

### 2.2. Study Population

This study was conducted in the Department of Emergency Medicine at Eunpyeong St. Mary’s Hospital, an 830-bed teaching hospital. The ED can accommodate approximately 50,000 patients annually. A physician provided the initial emergency treatment to all adult patients, defined as those older than 18 years, who visited the ED due to any medical problems between April 2019 and March 2022. We identified adult patients with urinary tract infection based on the discharge diagnosis. Among these patients, those with available laboratory and radiological abdominal computed tomography (CT) data were included in the study, and their medical records were reviewed to confirm the diagnosis of UTI. UTI diagnosis was confirmed by any International Classification of Disease 10th Edition’s ICD-10 (ICD-10) codes recorded in outpatient or admission medical documents as follows: urinary tract infection (N39.0), acute cystitis/acute cystitis without hematuria (N30.0), acute cystitis with hematuria (N30.01), other cystitis without hematuria (N30.8), other cystitis with hematuria (N30.81), unspecified cystitis without hematuria (N30.9), unspecified cystitis with hematuria (N30.91), and acute pyelonephritis (N10). Patients who were immunocompromised due to certain disease conditions, were transferred from another hospital, had indwelling urinary catheters, were pronounced dead upon arrival, or had received visit-irrelevant medical treatment were excluded.

### 2.3. Data Collection

We obtained the following demographic and clinical data from the medical records of the study participants: age, sex, and comorbidities, including malignancy, hypertension, diabetes mellitus (DM), coronary artery disease, cerebrovascular accident, congestive heart failure, chronic kidney disease, and pulmonary disease. Data on serum inflammatory biomarkers, such as high-sensitivity C-reactive protein level, erythrocyte sedimentation rate, and white blood cell count, were obtained during their time in the ED. The patients were categorized as alert, verbally responsive, pain responsive, or unresponsive.

### 2.4. Skeletal Muscle Index

We selected a single-cross-sectional image at the level of the L3 vertebral body from abdominal CT and measured the SMI with a touchscreen tablet and a stylus pen. We utilized ImageJ (https://imagej.nih.gov/ij/, (accessed on 1 January 2023)), free public-domain software developed by the National Institutes of Health [5,6]. A detailed description of the essential tools needed to use ImageJ software is described, followed by a step-by-step guide for analyzing a single diagnostic cross-sectional CT image (slice) with this program. We calculated the skeletal muscle area (SMA) by taking Measurement 3 (inner area, mm^2^), subtracting it from Measurement 2 (outer area mm^2^), and dividing the result by 100 in this program. Then, SMI was computed as SMA divided by height squared. Two investigators who were blinded to the clinical data independently assessed the CT scans and determined the SMI of all patients (Figure 1). The average SMI value between the two investigators was used.

### 2.5. Outcome Variables

The primary outcome was a critical event, defined as the initiation of dialysis, invasive ventilation, initiation of vasoactive medications, cardiac arrest, or death. The secondary outcomes were in-hospital mortality; ICU admission; length of hospital stay, which was how long the patient was at the hospital; and hospital admission, which was whether the patient was admitted to the hospital.

### 2.6. Statistical Analysis

Categorical variables are given as frequencies and percentages. Comparisons of categorical variables were performed using the χ2 test or Fisher’s exact test, as appropriate. We tested the distributions of the continuous variables for normality using visual inspection and the Shapiro–Wilk test. Normally distributed data are expressed as the means and standard deviations, and these data were assessed using Student’s *t*-test. Non-normally distributed data were assessed using the Mann–Whitney U test and expressed as the median and interquartile range. Sex-specific cutoff values for SMI at the L3 level measured by CT imaging in a healthy Korean population were used in this study [7]. Multivariate binary logistic regression analysis was used to assess independent predictors of critical events. All variables with a significance level < 0.1 by univariate analysis were included in a multivariate logistic regression model. Receiver operating characteristic (ROC) curves for predicting critical events were plotted for each model, and the predictive accuracy of each model was determined by the area under the ROC curve (AUROC) and the 95% confidence interval (CI). We also created combined models using several logistic regressions first and then saved the predicted probabilities. Using this saved probability as an indicator, one may conduct the ROC analysis. All of the statistical analyses were performed using SPSS software, version 23.0 (IBM Corp., Armonk, NY, USA), and MedCalc 12.0 (MedCalc Software, Inc., Mariakerke, Belgium). Values of *p* < 0.05 were considered statistically significant for all comparisons.

## 3. Results

### 3.1. Patient Characteristics

During the study period, a total of 478 patients who met the diagnostic criteria of a UTI and underwent an abdominal CT were included in this study. The patients were divided into two groups: those with a low SMI (n = 93) and those with a high SMI (n = 385). Table 1 indicates that patients in the low-SMI group were predominantly male (50.5%) and older (mean age of 73.3 years) than those in the high-SMI group. Patients with a low SMI also had a lower body mass index (BMI) than those with a high SMI. Patients with a low SMI were more likely to have hypertension and diabetes mellitus than those with a high SMI. In terms of the clinical findings upon admission, patients with a low SMI are more likely to have an altered mentality and higher qSOFA scores (an indicator of sepsis severity) than those with a high SMI. Regarding the laboratory findings, patients with a low SMI had significantly higher blood urea nitrogen (BUN) and creatinine levels, indicating poorer kidney function, than those with a high SMI. Patients with a low SMI tended to have higher body temperatures than those with a high SMI, but there was no significant difference in the bacteriuria rates between the two groups (Table 1).

### 3.2. Clinical Outcomes

In terms of hospital course, the high-SMI group had significantly lower in-hospital mortality (2.3% vs. 8.6%, *p* = 0.009), lower rates of ICU admission (7.8% vs. 28.0%, *p* < 0.001), and lower rates of critical events (13.8% vs. 50.5%, *p* < 0.001) than the low-SMI group. There was no significant difference in hospital admission rates between the two groups (57.1% vs. 64.5%, *p* = 0.239). Regarding specific interventions, the high-SMI group had significantly lower rates of dialysis (1.3% vs. 10.8%, *p* < 0.001), use of inotropics (7.5% vs. 29.0%, *p* < 0.001), refractory shock (8.3% vs. 34.4%, *p* < 0.001), and mechanical ventilation (1.0% vs. 8.6%, *p* < 0.001) than the low-SMI group. The high-SMI group had a significantly higher rate of weaning failure from mechanical ventilation than the low-SMI group (75.0% vs. 62.5%, *p* = 0.010). Additionally, the high-SMI group had a significantly shorter hospital stay (5.0 vs. 8.0 days, *p* < 0.001), while there was no significant difference in the length of ICU stay between the two groups (4.5 days for both groups, *p* = 0.560) (Table 2).

Figure 2 presents a comparison of the SMI values according to clinical outcomes, stratified by sex. For all clinical outcomes, there were significant differences according to the SMI. For males, there were significant differences in survival and the occurrence of critical events, while for females, there were significant differences in hospital admission, ICU admission, and the occurrence of critical events.

### 3.3. In-Hospital Mortality

Only 17 patients (3.6%) in the study died. Table 3 shows that the mean age of the nonsurvivors was higher than that of the survivors (78.8 ± 13.9 vs. 61.0 ± 19.7, *p* < 0.001). The nonsurvivors also had a lower BMI than the survivors (20.1 ± 4.7 vs. 23.5 ± 4.8, *p* = 0.005). In terms of premorbid diseases, a higher proportion of the nonsurvivors had diabetes mellitus and coronary artery disease (76.5% and 35.3%, *p* < 0.001 and *p* = 0.001, respectively) than the survivors. The difference in the proportion of patients with hypertension was not statistically significant (47.1% vs. 36.7%, *p* = 0.383). On admission, a higher proportion of the nonsurvivors than the survivors had an altered mentality (23.5% vs. 8.5%, *p* = 0.033) and qSOFA scores above 2 points (11.8% vs. 2.2%, *p* = 0.013). Among the laboratory findings, the nonsurvivors had higher levels of BUN (31.7 vs. 15.2, *p* = 0.010), creatinine (1.2 vs. 0.8, *p* < 0.001), and lactate (4.7 vs. 1.5, *p* < 0.001) than the survivors. The nonsurvivors also had a lower pH than the survivors (7.38 vs. 7.43, *p* = 0.037). Regarding the hospital course, a higher proportion of the nonsurvivors were admitted to the ICU than the survivors (76.5% vs. 9.3%, *p* < 0.001). The nonsurvivors had a longer length of ICU stay than the survivors, but the difference was not statistically significant (8.0 vs. 4.0, *p* = 0.137). The nonsurvivors also had a higher rate of critical events such as dialysis, use of inotropic, and refractory shock than the survivors (*p* < 0.001 for all). Moreover, the nonsurvivors had a higher rate of mechanical ventilation than the survivors (47.1% vs. 0.9%, *p* < 0.001).

### 3.4. Analysis for Predicting of Critical Events

The univariate analysis showed that male sex, older age, a lower BMI, a low SMI, and the presence of various pre-existing medical conditions were associated with increased odds of experiencing a critical event. The odds ratio of the low-L3-SMI group was 6.40 (95% CI: 3.90 to 10.59, *p* < 0.001). An altered mentality on admission, a higher respiratory rate, higher lactate levels, and a qSOFA score above 2 points were also strong predictors of critical events. On the other hand, a higher temperature and pH were associated with lower odds of experiencing a critical event. Bacteriuria was not a significant predictor of critical events.

Table 4 presents the results of multivariate logistic regression analysis for predicting critical events in urinary tract infection patients, with a focus on variables associated with the skeletal muscle index.

In model I, a low SMI, male sex, older age, and a higher BMI were included as independent variables. The adjusted odds ratio (aOR) for a low SMI was 3.09 (95% CI: 1.72–5.56), indicating that patients with a low SMI were more than three times as likely to experience critical events than those with a high SMI. The aOR for the male sex was 1.22 (95% CI: 0.70–2.13), which was not statistically significant. However, older age and a higher BMI were significantly associated with critical events, with aORs of 1.03 (95% CI: 1.02–1.05) and 0.91 (95% CI: 0.85–0.97), respectively.

In model II, a low SMI, diabetes mellitus, an altered mentality, lactate levels, and creatinine levels were included as independent variables, as they were found to be significant in univariate analysis. The aOR for a low SMI was 2.87 (95% CI: 1.10–7.53), indicating that patients with a low SMI were almost three times as likely to experience critical events as those with a high SMI, even after adjusting for other variables. In addition to a low SMI, diabetes mellitus, an altered mentality, lactate levels, and creatinine levels were all identified as significant predictors of critical events. The aORs for those were 2.81 (95% CI: 1.15–7.25), 8.15 (95% CI: 1.36–48.9), 1.45 (95% CI: 1.05–1.99), and 2.02 (95% CI: 1.03–3.96), respectively.

### 3.5. Prognostic Value of Model

In receiver operating characteristic (ROC) curve analysis, the area under the curve (AUC) of crude model I was 0.765 (95% CI 0.725–0.803, *p* < 0.001). And the AUC of crude model II was 0.833 (95% CI 0.754–0.895, *p* < 0.001). Model II had a sensitivity of 72% (95% CI 57.5–83.8) and a specificity of 80.0% (95% CI 68.7–88.6), while model I had a sensitivity of 75% (95% CI 65.3–83.1) and a specificity of 67.2% (95% CI 62.2–71.9).

## 4. Discussion

This is the first report about the association between critical outcome and the SMI in UTI patients. UTIs are a major cause of fever among patients presenting to the emergency department and can rapidly progress to sepsis, a life-threatening condition. In our study, we aimed to evaluate the contemporary annual burden of emergency department visits for UTIs. Our results revealed that almost 60% of patients diagnosed with UTIs in the emergency department required hospital admission, one-third of whom had a critical event while there. These findings underscore the potential severity of UTIs and their association with sepsis [8,9,10,11,12]. The scientific literature has consistently reported UTIs as a leading cause of fever among patients visiting emergency departments, and their potential to progress rapidly to sepsis is well-documented. Therefore, health care providers and policy makers need to recognize the significant health and economic burdens associated with UTIs and explore effective preventative measures and treatment options to improve outcomes.

UTIs are a common bacterial infection that can affect various parts of the urinary tract, including the bladder, urethra, and kidneys. Prognostic factors for UTIs refer to variables that can affect the course and outcome of the infection. Age is an essential factor, as UTIs are more common in women and the elderly. Comorbidities such as diabetes or kidney disease can increase the risk of complications from UTIs, and the severity of symptoms, including fever or nausea, can also affect the prognosis. The type of bacteria causing the infection is another important factor, as drug-resistant strains can lead to more severe infections and poorer outcomes. Patients who respond well to initial treatment are more likely to have a better prognosis. Recurrent UTIs can lead to more severe infections and complications, which can worsen the overall prognosis. Finally, the presence of complications such as kidney infections or sepsis can increase the risk of morbidity and mortality associated with UTIs.

Complicated urinary tract infections are a significant public health problem due to their high prevalence and impact on health care costs and patient outcomes. In recent years, there has been an increasing focus on complicated UTIs, and several studies were conducted to identify effective prevention and treatment strategies. One prospective multicenter study aimed to identify risk factors for complicated UTIs in hospitalized patients. The authors found that older age, male sex, a previous UTI, and comorbidities such as diabetes and chronic kidney disease were independent risk factors for complicated UTIs. They suggested that these factors should be considered when developing strategies to prevent and manage complicated UTIs in hospitalized patients [13]. Another review article discussed the challenges of diagnosing and treating complicated UTIs and offered potential solutions, such as using rapid diagnostic tests, optimizing antibiotic therapy, and implementing antimicrobial stewardship programs. The authors also highlighted the need for further research to better understand the epidemiology and pathogenesis of complicated UTIs and to develop more effective treatment strategies [14]. Overall, these studies conducted on complicated UTIs highlight the significant impact of complicated UTIs on both patient outcomes and health care costs. Since cUTIs have unfavorable social and economic consequences, studies that examine prognostic factors for UTIs, such as ICU admission and the occurrence of critical events, are crucial.

Several studies investigated the correlation between the skeletal muscle index and outcomes in critical care patients. The SMI is a measure of muscle mass indexed to height, and it is typically assessed using imaging modalities such as computed tomography or magnetic resonance imaging (MRI). The findings of multiple studies suggested that SMI may be a valuable tool for predicting clinical outcomes and identifying patients who may benefit from early rehabilitation interventions in critical care settings [15,16,17]. 

The skeletal muscle index is a measure of the amount of skeletal muscle mass present in the body, usually expressed in kilograms per square meter (kg/m^2^). Critically ill patients are particularly vulnerable to muscle wasting due to immobility, systemic inflammation, and catabolic processes associated with critical illness. The SMI can be measured using imaging techniques such as computed tomography and magnetic resonance imaging at different levels of the body, such as the mid-thigh or the third lumbar vertebrae. These imaging techniques allow for the assessment of the muscle cross-sectional area, which can then be used to calculate the SMI. There is no single best method for measuring the SMI in critically ill patients. Each imaging technique has its own advantages and limitations, and the choice of method depends on the available resources, patient condition, and purpose of measurement. 

Our study used CT to measure the SMI, which has the advantages of high interrater reliability and accuracy and provides additional diagnostic data. Other studies dis-cussed the use of different methods for measuring the SMI in critically ill patients. One study compared three different methods, CT, BIA, and an ultrasound, and found good agreement among the methods. This suggests that BIA and an ultrasound may be feasible alternatives to CT for measuring the SMI in critically ill patients [18,19,20,21]. In our previous study, the association between skeletal muscle mass and pneumonia severity in elderly patients visiting the emergency department was investigated. The L3 skeletal muscle index (L3SMI) was calculated by measuring skeletal muscle mass from computed tomography (CT) images of the third lumbar vertebrae. The study revealed that lower L3-SMI values were associated with a higher risk of severe pneumonia in elderly patients visiting the emergency department. We suggest that the L3-SMI is a noninvasive and simple tool that could be used to identify high-risk elderly patients with severe pneumonia [3].

Studies consistently showed that critically ill patients with a low skeletal muscle index (SMI) experience increased morbidity and mortality, prolonged hospital stays, and decreased physical function after discharge. Nagahama et al. conducted a prospective cohort study that found a strong association between a lower SMI and an increased risk of all-cause mortality. After adjusting for various factors, including age, lifestyle, and medical history, the study reported a hazard ratio of 3.72 (95% CI, 1.73–7.99) for all-cause mortality in the lowest tertile of the SMI compared to the highest tertile. These findings underscore the importance of the SMI as a significant predictor of all-cause mortality, especially in critical care settings [22].

SMI was also studied as a prognostic factor in various disease groups, such as cancer, liver cirrhosis, chronic obstructive pulmonary disease (COPD), and heart failure. For instance, in patients with various types of cancer, including lung, pancreatic, and colorectal cancer, a low SMI was associated with a poor prognosis. Similarly, a decreased SMI was linked to a higher risk of complications and mortality in patients with liver cirrhosis. In patients with COPD, a low SMI was associated with an increased risk of mortality and hospitalization. Additionally, a reduced SMI was found to be associated with a higher risk of morbidity and mortality in patients with heart failure [23,24,25,26,27,28].

The ability to track changes in muscle mass over time and assess the effectiveness of interventions, such as nutrition and physical therapy, makes the SMI a valuable tool in critical care settings. Identifying muscle wasting in critically ill patients through SMI measurements provides critical information on patient outcomes and the effectiveness of interventions aimed at preserving muscle mass. The variables identified in our study, including a low SMI, diabetes mellitus, an altered mental state, lactate levels, and creatinine levels, are all readily available and easily measurable during an initial diagnosis in the emergency department. Therefore, our findings suggest a potentially useful initial prognostic model for UTI patients.

Incorporating the SMI into the initial prognostic model for UTI patients may have significant clinical implications. The early identification of patients at a high risk for adverse outcomes could allow prompt and appropriate treatment, potentially improving outcomes and reducing health care costs. Furthermore, the noninvasive nature of SMI measurement through imaging modalities like CT makes it a valuable prognostic tool without the need for more invasive testing. Future studies are needed to validate and refine this initial prognostic model for UTIs and evaluate its clinical impact on patient outcomes.

Given the close relationship between the SMI and hospital course, health care providers can utilize SMI information as a valuable tool to optimize treatment decisions, resulting in improved patient outcomes and the more efficient use of health care resources. Furthermore, these results may inform future research and clinical practices in the field of UTI diagnosis and treatment. As UTIs are common infections that can lead to serious health complications, including sepsis, it is crucial to address the impact of muscle wasting on patient outcomes. The early identification of muscle wasting through SMI measurement can facilitate an early intervention, such as nutritional support, physical therapy, and early mobilization, to improve a patient’s nutritional status and prevent further muscle loss. Addressing these factors can improve outcomes and reduce health care costs in patients with UTIs.

There are several limitations to the present study that need to be acknowledged. First, this study’s retrospective design precludes determining causal relationships between the variables that were studied. It is possible that other factors that were not accounted for in the analysis may have influenced the observed associations. Second, the study’s sample size was limited to a specific geographic region, thereby possibly limiting its generalizability to other populations. Third, the study relied on self-reported data, which may have introduced recall bias or social desirability bias. Moreover, the study did not comprehensively assess all the potential risk factors for the health outcomes that were studied, such as family history, diet, and lifestyle factors. Last, the study did not collect information on the severity or duration of health conditions, which may have impacted the associations that were observed.

## 5. Conclusions

The gender-specific low SMI is an independent factor associated with the occurrence of critical events in UTI patients during hospitalization. Additionally, there were more critical events in patients who died in the hospital. When developing guidelines or protocols for monitoring and managing UTI patients with a low SMI, the patient’s SMI may be considered along with other clinical factors and medical history, which may include regular assessments of nutritional status, monitoring for signs of infection, and implementing interventions to prevent the development of critical events. 

## Figures and Tables

**Figure 1 jcm-12-05024-f001:**
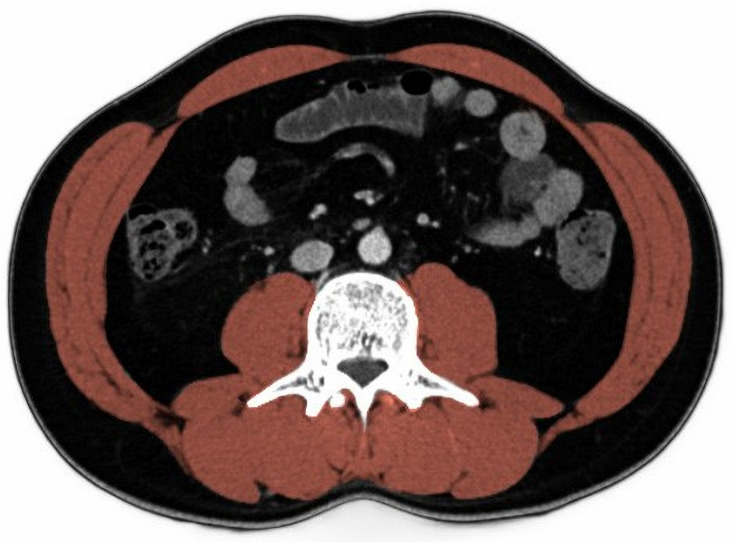
Representative CT images of L3 skeletal muscle area measurement by ImageJ: total skeletal muscle area was measured using the following four components: paraspinal muscle, extracostal abdominal wall muscle, intercostal muscle, and psoas muscle and diaphragm.

**Figure 2 jcm-12-05024-f002:**
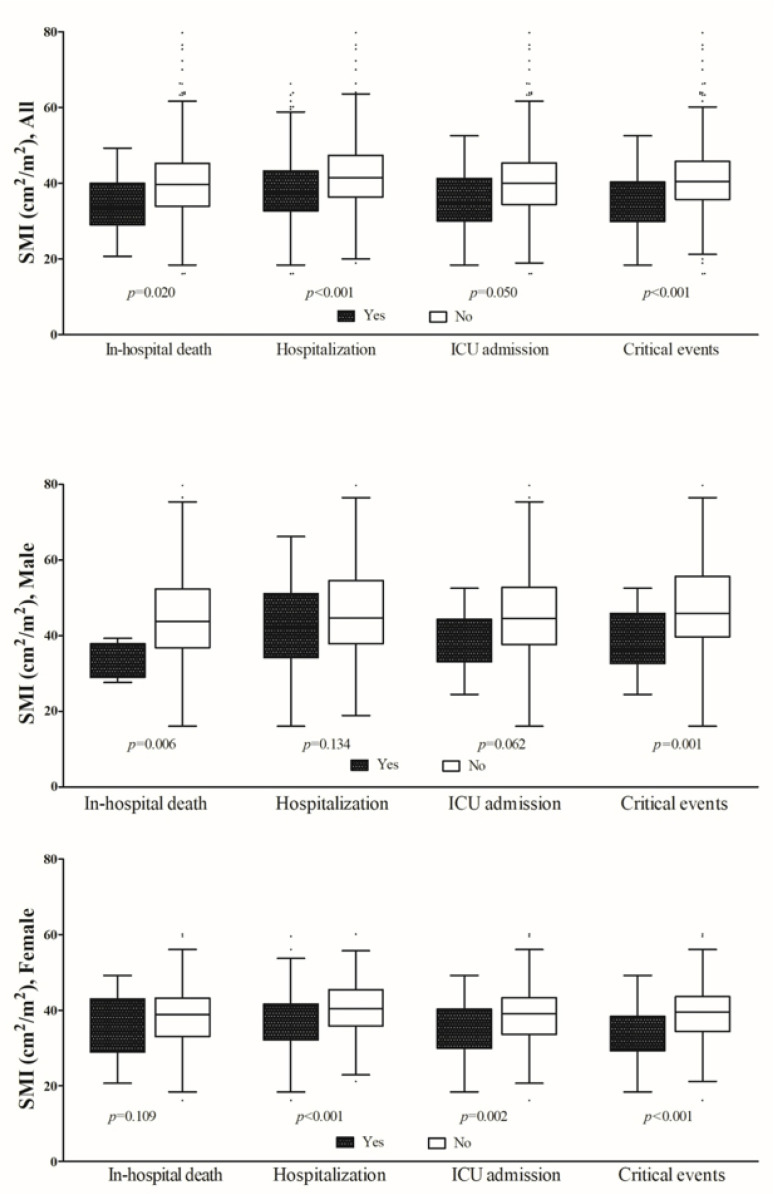
Comparison of SMI values according to clinical outcomes stratified by gender.

**Table 1 jcm-12-05024-t001:** Baseline characteristics of patients with urinary tract infection by SMI.

	All	Low SMI	High SMI	*p*
	N = 478	n = 93	n = 385	
Male, n (%)	120 (25.1)	47 (50.5)	73 (19.0)	<0.001
Age (mean ± SD)	61.6 ± 19.8	73.3 ± 17.6	58.8 ± 19.3	<0.001
BMI	23.4 ± 4.9	20.4 ± 4.9	24.1 ± 4.6	<0.001
SMI (cm^2^/m^2^)				
Male	45.4 ± 13.4	33.5 ± 5.5	53.0 ± 11.2	
Female	38.5 ± 7.3	26.9 ± 3.2	40.2 ± 6.1	
Premorbid disease, n (%)				
Hypertension	177 (37.0)	51 (54.8)	126 (32.7)	<0.001
Diabetes mellitus	143 (29.9)	40 (43.0)	103 (26.8)	0.003
Coronary artery disease	52 (10.9)	11 (11.8)	41 (10.6)	0.887
Cerebrovascular accident	41 (8.6)	14 (15.1)	27 (7.0)	0.023
Pulmonary disease	14 (2.9)	5 (5.4)	9 (2.3)	0.161
Chronic kidney disease	21 (4.4)	7 (7.5)	14 (3.6)	0.152
Clinical findings on admission				
Altered mentality, n (%)	43 (9.0)	13 (14.0)	30 (7.8)	0.095
Vital signs (mean ± SD)				
Mean arterial pressure, mmHg	93.1 ± 17.2	94.6 ± 21.6	92.7 ± 15.9	0.429
Heart rate/min	90.6 ± 19.1	93.6 ± 21.1	89.8 ± 18.5	0.083
Respiratory rate/min	19.0 ± 2.8	19.4 ± 3.4	18.9 ± 2.6	0.211
Temperature, °C	37.6 ± 1.2	37.2 ± 1.1	37.7 ± 1.2	0.001
qSOFA above 2 points	12 (2.5)	7 (7.6)	5 (1.3)	0.001
Laboratory findings, median (IQR)				
White blood cells (×10^9^/L)	11.0 (8.1–14.8)	11.7 (7.9–15.5)	10.9 (8.1–14.6)	0.548
BUN (mg/dL)	15.5 (11.8–22.2)	22.2 (14.7–36.1)	14.5 (11.4–20.7)	<0.001
Creatinine (mg/dL)	0.8 (0.7–1.1)	0.9 (0.7–1.4)	0.8 (0.6–1.0)	0.005
C-reactive protein (mg/dL)	7.5 (1.4–15.2)	6.6 (1.6–13.4)	7.6 (1.4–15.4)	0.756
ESR (mm/h)	27.0 (10.0–49.0)	31.0 (9.0–52.0)	26.0 (10.0–49.0)	0.535
Lactate, mmol/L	1.6 (1.1–2.2)	1.8 (1.2–2.6)	1.5 (1.1–2.0)	0.082
pH	7.43 (7.38–7.47)	7.42 (7.35–7.47)	7.43 (7.39–7.47)	0.367
Bacteriuria	285 (59.6%)	50 (53.8%)	235 (61.0%)	0.244

Values are means ± standard deviation or median (interquartile range) or numbers and percentages. BMI, body mass index; SMI, skeletal muscle index; qSOFA, quick Sepsis-related Organ Failure Assessment; ESR, erythrocyte sedimentation rate.

**Table 2 jcm-12-05024-t002:** Comparison of outcomes between low-SMI group and high-SMI group.

	All	Low SMI	High SMI	*p*
	N = 478	n = 93	n = 385	
Hospital courses				
In-hospital mortality	17 (3.6)	8 (8.6)	9 (2.3)	0.009
Hospitalization	280 (58.6)	60 (64.5)	220 (57.1)	0.239
ICU admission	56 (11.7)	26 (28.0)	30 (7.8)	<0.001
Length of ICU stay	4.5 (3.0–10.5)	4.5 (3.0–14.5)	4.5 (3.0–7.5)	0.560
Length of hospital stay	5.0 (1.0–9.0)	8.0 (2.0–14.0)	5.0 (1.0–8.0)	<0.001
Critical events, n (%)	100 (20.9)	47 (50.5)	53 (13.8)	<0.001
Dialysis	15 (3.1)	10 (10.8)	5 (1.3)	<0.001
Use of inotropic	56 (11.7)	27 (29.0)	29 (7.5)	<0.001
Refractory shock	64 (13.4)	32 (34.4)	32 (8.3)	<0.001
Mechanical ventilation	12 (2.5)	8 (8.6)	4 (1.0)	<0.001
Weaning failure, n = 12	8 (66.7)	5 (62.5)	3 (75.0)	0.010

Values are median (interquartile range) or numbers and percentages.

**Table 3 jcm-12-05024-t003:** Characteristics and hospital courses in patients with urinary tract infection by survival discharge.

	Nonsurvivors	Survivors	*p*
	n = 17	n = 461	
Male, n (%)	4 (23.5)	116 (25.2)	0.879
Age (mean ± SD)	78.8 ± 13.9	61.0 ± 19.7	<0.001
BMI	20.1 ± 4.7	23.5 ± 4.8	0.005
SMI (cm^2^/m^2^)			0.020
Male	33.4 ± 4.8	45.8 ± 13.4	
Female	35.3 ± 8.8	38.6 ± 7.6	
Premorbid disease, n (%)			
Hypertension	8 (47.1)	169 (36.7)	0.383
Diabetes mellitus	13 (76.5)	130 (28.2)	<0.001
Coronary artery disease	6 (35.3)	46 (10.0)	0.001
Cerebrovascular accident	1 (5.9)	40 (8.7)	0.686
Pulmonary disease	0 (0.0)	14 (3.0)	0.466
Chronic kidney disease	1 (5.9)	20 (4.3)	0.760
Clinical findings on admission			
Altered mentality, n (%)	4 (23.5)	39 (8.5)	0.033
Vital signs (mean ± SD)			
Mean arterial pressure, mmHg	93.2 ± 23.1	93.1 ± 16.9	0.986
Heart rate/min	88.7 ± 16.0	90.6 ± 19.2	0.674
Respiratory rate/min	19.8 ± 3.5	19.0 ± 2.8	0.258
Temperature, °C	36.6 ± 1.2	37.6 ± 1.2	<0.001
qSOFA above 2 points	2 (11.8)	10 (2.2)	0.013
Laboratory finding, median (IQR)			
White blood cells (×10^9^/L)	12.2 (8.3–17.4)	11.0 (8.1–14.7)	0.127
BUN (mg/dl)	31.7 (24.8–50.9)	15.2 (11.7–21.8)	0.010
Creatinine (mg/dl)	1.2 (0.9–1.8)	0.8 (0.7–1.1)	<0.001
C-reactive protein (mg/dL)	5.0 (2.3–18.3)	7.6 (1.4–15.1)	0.722
ESR (mm/h)	24.0 (8.0–60.0)	27.0 (10.0–49.0)	0.763
Lactate, mmol/L	4.7 (1.9–7.0)	1.5 (1.1–2.0)	<0.001
pH	7.38 (7.06–7.45)	7.43 (7.39–7.47)	0.037
Bacteriuria	13 (76.5)	272 (59.0)	0.149
Hospital course			
ICU admission	13 (76.5)	43 (9.3)	<0.001
Length of ICU stay	8.0 (4.0–14.0)	4.0 (3.0–9.0)	0.137
Length of hospital stay	7.0 (3.0–11.0)	5.0 (1.0–9.0)	0.411
Critical events, n (%)	17 (100.0)	83 (18.0)	<0.001
Dialysis	6 (35.3)	9 (2.0)	<0.001
Use of inotropic	16 (94.1)	40 (8.7)	<0.001
Refractory shock	17 (100.0)	47 (10.2)	<0.001
Mechanical ventilation	8 (47.1)	4 (0.9)	<0.001

Values are means ± standard deviation or median (interquartile range) or numbers and percentages. BMI, body mass index; SMI, skeletal muscle index; qSOFA, quick Sepsis-related Organ Failure Assessment; ESR, erythrocyte sedimentation rate.

**Table 4 jcm-12-05024-t004:** Multivariate logistic regression analysis for predicting critical events, with a focus on variables associated with skeletal muscle index.

	aOR (95% CI)	*p*
Model I (adjusted for gender, age, BMI)		
Low SMI	3.09 (1.72–5.56)	<0.001
Male	1.22 (0.70–2.13)	0.473
Age	1.03 (1.02–1.05)	<0.001
BMI	0.91 (0.85–0.97)	0.003
Model II		
Low SMI	2.87 (1.10–7.53)	0.031
Diabetes mellitus	2.81 (1.15–7.25)	0.025
Altered mentality	8.15 (1.36–48.9)	0.022
Lactate, mmol/L	1.45 (1.05–1.99)	0.024
Creatinine (mg/dl)	2.02 (1.03–3.96)	0.041

## Data Availability

Due to privacy restrictions, all data are stored at the Corresponding authorinstitution. Qualified researchers are able to gain access via an application to the corresponding author.

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
