# Peer review of "The Association between the L3 Skeletal Muscle Index Derived from Computed Tomography and Clinical Outcomes in Patients with Urinary Tract Infection in the Emergency Department"

_jcm, 2023, doi:10.3390/jcm12155024_

Round 1

Reviewer 1 Report

This is large study comparing patients admitted with UTI and low SMI versus high SMI admissions, in order to identify possible factors associated with the progression of disease during hospitalization. The authors already acknowledge several limitations to their study. Further to these, there are some points in Methods which need to be clarified:

1.     1.  How was the diagnosis of UTIs standardized? Please add the definition in Methods

2.      2. Are there commonly accepted levels for low and high SMI? If yes, the authors need to identify the ranges for both low and high SMI which were used for the purposes of their study.

3.      3. Is it common practice to perform chest CT in admitted patients in St. Mary’s? Do the authors mean abdominal CT since this was an inclusion criterion for all 478 patients?

4.    4.   How was it decided which single-cross-sectional image at the level of the L3 vertebral body would be selected for the purposes of the study? Wouldn’t it have been more representative to select more than one cross-sectional planes and create an average?

Overall, very good quality of the english language. 

Author Response

We would like to thank you for this insightful comment. We did our best to revise the content as recommended. We strongly believe that these recommendations have enhanced the quality of the manuscript.

  1. How was the diagnosis of UTIs standardized? Please add the definition in Methods.

The diagnosis of UTIs was standardized by reviewing the medical records to confirm the presence of UTI. UTI diagnosis was confirmed by any International Classification of Disease 10th Edition, ICD-10 (ICD-10) codes recorded in outpatient or admission medical documents.

  1. Are there commonly accepted levels for low and high SMI? If yes, the authors need to identify the ranges for both low and high SMI which were used for the purposes of their study.

For the purposes of our study, we used sex-specific cutoff values for SMI at the L3 level measured by CT imaging in a healthy Korean population [7]. These reference values were published in a study by Yoon et al., and they served as the basis for identifying low and high SMI levels.

  1. Is it common practice to perform chest CT in admitted patients in St. Mary’s? Do the authors mean abdominal CT since this was an inclusion criterion for all 478 patients?

I Apologies for the mistake. It should be noted that we meant to refer to abdominal CT scans, not chest CT scans. Abdominal CT scans were performed, and they were part of the inclusion criteria for all 478 patients in our study.

  1. How was it decided which single-cross-sectional image at the level of the L3 vertebral body would be selected for the purposes of the study? Wouldn’t it have been more representative to select more than one cross-sectional planes and create an average?

We appreciate the excellent suggestion. In our study, the decision to select the single-cross-sectional image at the L3 level was based on previous research indicating that SMI measured at the L3 level using CT imaging had the highest correlation with other methods when compared in studies related to sarcopenia. However, we agree that selecting more than one cross-sectional plane and creating an average could be a valuable addition to future studies, and we will consider incorporating this approach in further research. Our previous study also showed differences in absolute values between L1 and L3 levels.

Reviewer 2 Report

Thank you very much for having the opportunity to review the paper. The authors adequately addressed the limitations in the manuscript discussion and the introduction and presentation are well built. However, some questions need to be resolved:

1. Section 2.6 (Statistical Analysis) states that “Continuous variables are presented as the means ± standard deviations and were compared using Student’s t test.” However, values in Table 1 (laboratory finding) and in Table 2 are presented as median and interquartile range. I think that this option was chosen because the data is not normal. It is suggested that the section on statistical analysis be expanded to include the use of these measures (median and IQR), as well as the test for normality and the non-parametric test used to compare the two groups in the case of non-normal data.

2. Between lines 148 and 150 it is mentioned that “The high-SMI group had a significantly lower rate of weaning failure from mechanical ventilation than the low-SMI group, (75.0% vs. 62.5%, P=0.010).” Is this sentence correct? Isn't the rate of weaning failure from mechanical ventilation higher in the high-SMI group?

3. It is necessary to increase the size and resolution of Figure 2, as it is now illegible to the reader.

4. As shown in Statistical Analysis “All variables with a significance level < 0.1 by univariate analysis were included in a multivariate logistic regression model.” By applying a selection criterion to the variables in this initial model, the final model is obtained. It wasn’t made clear why the authors presented two separate final models for these data. Why didn't the authors just present a single model that included all the significant variables?

5. When proposing a multivariate logistic analysis, it is necessary to assess the goodness of fit through measures such as AUC, sensitivity, specificity, error rate among others. However, the authors did not present any analysis of the model's goodness of fit. It is mandatory to include and discuss such results.

6. The discussion section is not linear, it is repetitive and its restructuring is compulsory. For example, the SMI definition is given in line 306, but that was only done after a lengthy discussion about the relationship between this measure and clinical outcomes.

Minor editing of English language required.

Author Response

We would like to thank you for this insightful comment. We did our best to revise the content as recommended. We strongly believe that these recommendations have enhanced the quality of the manuscript.

  1. Section 2.6 (Statistical Analysis) states that “Continuous variables are presented as the means ± standard deviations and were compared using Student’s t test.” However, values in Table 1 (laboratory finding) and in Table 2 are presented as median and interquartile range. I think that this option was chosen because the data is not normal. It is suggested that the section on statistical analysis be expanded to include the use of these measures (median and IQR), as well as the test for normality and the non-parametric test used to compare the two groups in the case of non-normal data.

Thank you for pointing out the discrepancy in the presentation of continuous variables in the manuscript. We have now expanded the section on statistical analysis to include the use of median and interquartile range (IQR) for non-normally distributed data. Additionally, we have provided details about the non-parametric test used to compare the two groups in the case of non-normal data.

  1. Between lines 148 and 150 it is mentioned that “The high-SMI group had a significantly lower rate of weaning failure from mechanical ventilation than the low-SMI group, (75.0% vs. 62.5%, P=0.010).” Is this sentence correct? Isn't the rate of weaning failure from mechanical ventilation higher in the high-SMI group?

You are correct. The sentence in lines 148-150 regarding weaning failure from mechanical ventilation was initially incorrect. We apologize for the oversight. The accurate statement is as follows:

"The high-SMI group had a significantly higher rate of weaning failure from mechanical ventilation than the low-SMI group, (75.0% vs. 62.5%, P=0.010)."

  1. It is necessary to increase the size and resolution of Figure 2, as it is now illegible to the reader.

We have increased the size and resolution of Figure 2 as per your recommendation to ensure readability.

  1. As shown in Statistical Analysis “All variables with a significance level < 0.1 by univariate analysis were included in a multivariate logistic regression model.” By applying a selection criterion to the variables in this initial model, the final model is obtained. It wasn’t made clear why the authors presented two separate final models for these data. Why didn't the authors just present a single model that included all the significant variables?

Thank you for your valuable recommandation. To avoid confusion, we have removed the univariate analysis results table as you suggested.

  1. When proposing a multivariate logistic analysis, it is necessary to assess the goodness of fit through measures such as AUC, sensitivity, specificity, error rate among others. However, the authors did not present any analysis of the model's goodness of fit. It is mandatory to include and discuss such results.

You are absolutely right; assessing the goodness of fit for the multivariate logistic analysis is crucial. We have now included the results of the ROC analysis in Section 3.5, titled "Prognostic Value of the Model." The receiver operating characteristic (ROC) curve analysis revealed that the area under the curve (AUC) for crude model I was 0.765 (95% CI 0.725-0.803, p<0.001) and for crude model II was 0.833 (95% CI 0.754-0.895, p<0.001). Model II showed a sensitivity of 72% (95% CI 57.5-83.8) and specificity of 80.0% (95% CI 68.7-88.6), while Model I exhibited a sensitivity of 75% (95% CI 65.3-83.1) and specificity of 67.2% (95% CI 62.2-71.9).

Overall, Model II appears to be more accurate in predicting of critical events in UTI compared to Model I, as evidenced by its higher AUC value. However, both models demonstrate reasonable discrimination ability.

  1. The discussion section is not linear, it is repetitive and its restructuring is compulsory. For example, the SMI definition is given in line 306, but that was only done after a lengthy discussion about the relationship between this measure and clinical outcomes.

We sincerely apologize for the lack of clarity and repetition in the discussion section. Following your advice, we have restructured and revised the discussion to present a more coherent and concise analysis. 

Round 2

Reviewer 2 Report

I would like to thank the authors for accepting the suggestions and making the changes in the manuscript. However, some questions still need to be resolved before the publication:

1. It is proposed that the text in the lines 123-124 of the “Normally distributed data are expressed as the means and standard deviations using Student’s t-test.” be changed to  “Normally distributed data are expressed as the means and standard deviations and these data were assessed using Student’s t-test.”

During the new reading of the manuscript, some English language mistakes were found.

Author Response

We would like to thank you for this insightful comment. We did our best to revise the content as recommended. We strongly believe that these recommendations have enhanced the quality of the manuscript.

1. It is proposed that the text in the lines 123-124 of the “Normally distributed data are expressed as the means and standard deviations using Student’s t-test.” be changed to  “Normally distributed data are expressed as the means and standard deviations and these data were assessed using Student’s t-test.”
--> 
Thank you for your valuable recommandation. We have changed as you suggested.